# Universally Immune: How Infection Permissive Next Generation Influenza Vaccines May Affect Population Immunity and Viral Spread

**DOI:** 10.3390/v13091779

**Published:** 2021-09-06

**Authors:** Maireid B. Bull, Carolyn A. Cohen, Nancy H.L. Leung, Sophie A. Valkenburg

**Affiliations:** 1HKU-Pasteur Research Pole, School of Public Health, The University of Hong Kong, Hong Kong, China; maireid@connect.hku.hk (M.B.B.); ccohen@connect.hku.hk (C.A.C.); 2World Health Organization Collaborating Centre for Infectious Disease Epidemiology and Control, School of Public Health, The University of Hong Kong, Hong Kong, China; leungnan@hku.hk

**Keywords:** influenza, vaccine, T cell, immune escape, prior immunity

## Abstract

Next generation influenza vaccines that target conserved epitopes are becoming a clinical reality but still have challenges to overcome. Universal next generation vaccines are considered a vital tool to combat future pandemic viruses and have the potential to vastly improve long-term protection against seasonal influenza viruses. Key vaccine strategies include HA-stem and T cell activating vaccines; however, they could have unintended effects for virus adaptation as they recognise the virus after cell entry and do not directly block infection. This may lead to immune pressure on residual viruses. The potential for immune escape is already evident, for both the HA stem and T cell epitopes, and mosaic approaches for pre-emptive immune priming may be needed to circumvent key variants. Live attenuated influenza vaccines have not been immunogenic enough to boost T cells in adults with established prior immunity. Therefore, viral vectors or peptide approaches are key to harnessing T cell responses. A plethora of viral vector vaccines and routes of administration may be needed for next generation vaccine strategies that require repeated long-term administration to overcome vector immunity and increase our arsenal against diverse influenza viruses.

## 1. Introduction

Seasonal influenza viruses contribute to a substantial disease burden globally and estimated to result in 290,000–650,000 deaths annually, while pandemics can have an even greater impact [1]. The first influenza vaccine was developed in 1936, against influenza A virus (IAV), and the standard format of inactivated vaccines in widespread use remains the same 80 years later [2]. However, vaccinology continues to develop and improve. The ongoing COVID-19 pandemic has exemplified the benefits of vaccine preparedness, which would be especially important in the event of a novel IAV pandemic strain. The potential for highly pathogenic avian influenza (HPAI) viruses to acquire mutations to facilitate improved human-to-human transmission to cause pandemics [3,4] is a very real future threat with the potential to dwarf the impact caused by SARS-CoV-2.

Current influenza vaccination strategies primarily rely on inactivated influenza virus (IIV) vaccines for seasonal protection against IAV by eliciting antibody-mediated protection against the highly mutable surface protein haemagglutinin (HA), as virus neutralization by HA head-specific antibodies can block infection. This strategy relies on the successful prediction of future seasonal virus strains and minimal egg adaption during large-scale growth of vaccine strains in eggs. In 2020/2021, cases of seasonal influenza are at a historically low rate, decreasing from a prevalence of >20% in clinical and surveillance samples to just 2.3% in the US [5], indicating that public health measures to mitigate COVID-19 also impact the transmission of IAV. While this is a relative reprieve for hospital systems still dealing with an influx of COVID-19 patients, it reduces the capacity to accurately predict the next seasonal strains for incorporation into vaccines. As seasonal IIV does not provide protection for future seasonal or potentially arising pandemic virus strains, there has been a push to develop universal or next generation vaccines which could provide cross-reactive heterosubtypic immunity for different influenza strains and subtypes for an extended duration. Influenza A viruses have at least 17 different HA and 11 NA subtypes subject to continual drift, but the coverage of the universal vaccine targets vary by vaccine type. Heterosubtypic vaccines can be either pan-influenza covering both influenza A and B viruses, group-specific based on HA phylogeny (e.g., group 1 HA H1N1/H5N1 versus group 2 HA H3N2/H7N9), or subtype specific (different H1N1 strains) depending on the breadth of responses elicited and epitope conservation that is targeted by those vaccines. As part of the drive for universal IAV vaccine development, the National Institutes of Health set out a framework in 2018 for the goals of next generation universal vaccines [6]. Ideally a universal vaccine should protect against 75% of IAV strains, be 75% effective at preventing symptomatic influenza, be suitable for all age groups, and provide at least 1 year of protection.

Currently, there are a multitude of varying strategies and methods being used to develop a next generation vaccine for IAV pandemic preparedness [7], all of which aim to achieve heterosubtypic immunity to different influenza strains and subtypes in varying ways. They also may target different outcomes, such as reduction in the number of infections, risk of different disease outcomes after infection (e.g., hospitalisation or death) or propensity to transmit after infection [8,9]. IIVs confer protection by primarily eliciting specific responses towards the antigenically variable globular HA region; Whereas live attenuated influenza vaccines (LAIV) or novel next-generation vaccine strategies, such as chimeric HA stem vaccines, neuraminidase (NA)-based vaccines, peptide mosaic vaccines, and viral vectored vaccines (such as adenovirus- or vaccinia virus-based vectors) were designed to target conserved IAV regions and generate cross-reactive immune responses by targeting different stages of the virus life cycle. For example, HA stem antibodies can interfere with virus fusion after entry, whilst T cells are primed after limited replication of the vaccine strains or local antigen presentation. This review explores the interplay of each of these novel vaccine designs with prior immunity, and their potential for enhanced immune pressure and viral evasion due to non-sterilising functions.

## 2. Prior Immunity Can Impact Vaccine Responses

Previous influenza infections and vaccinations shape individual immunity and impact vaccine response [10,11]. Age is a primary determinant of vaccine responsiveness due to several factors, including immunosenescence in the elderly [12,13], differences in infection and vaccination history [14,15], persistent cytomegalovirus (CMV) infection [16], and a shift towards anti-inflammatory responses in children [17]. Older adults and young children are impacted in different ways by their prior immunity or lack thereof, and IIV responses may be shaped by previous immune priming [18,19,20]. This phenomenon can be explained by the theory of original antigenic sin (OAS), or imprinting, first described by Thomas Francis in 1960 [21] and extensively reviewed more recently [22,23,24]. OAS proposes that early exposure to influenza shapes the immune system’s ability to respond to future exposures, whether in the form of infection or vaccination, due to preferential recall of memory antibody responses against conserved regions instead of *de novo* activation of naïve B cells specific to new antigens [25]. Modelling studies from Gostic et al. show that within-subtype imprinting in early childhood can confer greater protection against other viruses of the same group, e.g., H1N1 for H5N1 and H3N2 for H7N7 infection [26], and this also impacted the severity of seasonal influenza viruses [27,28]. Therefore, in the context of next generation influenza vaccines, it is important to elicit cross-reactive heterosubtypic responses to both HA groups to operate within an immune population.

LAIV, an A/Ann Arbor backbone (FluMist, Fluenz) or Russian backbone (Ultravac and Nasovac-S), induces a broader immune response than IIV, with the inherent ability to elicit cross protective cellular immunity and potent mucosal IgA antibodies [29]. However, in a small study of Flumist in South Africa with over 60-year-olds, there is no increased rate of protection from acquisition of infection or duration of immunity for LAIV compared to IIV. The age of study participants (60–95 years old) and low incidence of infection may preclude these findings [30]. As children are relatively immunologically naïve subjects, the ability to boost T cell responses may be inversely proportional to their prior exposure. Some studies have shown good boosting in cross-protective CD8^+^ T cells [31], and CD4^+^ T cells after vaccination [32]. However, there are also some reports of lower vaccine efficacy in children (aged 2 to 17) [33] which may be attributable to a lack of cross-protective CD8^+^ T cell responses at the expense of a good antibody response [31]. Therefore, it is difficult with current available LAIV vaccines to boost both cross-reactive T cell responses and neutralising antibodies due to prior immunity. Whilst original antigenic sin may account for this phenomenon, it is also possible that the available strains for LAIV are too attenuated for efficient viral replication in the upper nasal passages to prime cellular immunity and high titre mucosal antibodies [34]. Furthermore, adults aged 50–64 were found to not be protected against vaccine mismatched influenza A viruses [35] or antigenically similar B virus [36] following LAIV compared to placebo. When prior influenza exposure is modelled in mice via intramuscular IIV priming, LAIV protects against challenge with the same homologous strain, with an absence of any cross-reactive antibody responses [37]. From this mouse study, Roy et al. suggested that eliciting cross-reactive T cells during infection were suppressed by pre-existing antibodies, though this was not specifically tested. Prior serological immunity without T cell priming, such as LAIV after IIV, may hamper the success of LAIV. Currently, IIV is licensed from 6 months of age and LAIV is only licensed from 2 years and older, which may currently limit the early development of influenza-specific T cell responses by vaccination. More promisingly, however, sequential LAIV vaccinations in ferrets confers improved protection against homologous and heterologous challenge [38]. This suggests that initially priming the immune system with a cellular immune response may improve post-vaccine immune responses and will have implications for the introduction of more widespread LAIV use from a young age.

## 3. T Cell Responses in the Context of Current Vaccines and Next Generation Design

Another strategy for developing a broadly reactive next generation vaccine to provide heterosubtypic protection is priming or boosting T cells, which have the ability to recognise conserved viral peptides across different influenza subtypes. T cells were found to have essential protective functions from severe disease [39], reduction of viral loads [40] and reduction of symptom duration and fever [41], especially in the absence of neutralizing antibodies where T cells exhibit remarkable cross-reactivity to novel strains. A modelling study involving a hypothetical T cell-inducing vaccine suggested that efficacy in the context of a novel pandemic will depend heavily on pre-existing immunity [42]. Bolton et al. predict that when attempting to ‘immunise the immune’, for example by administering a T cell-inducing vaccine to a population possessing cross-reactive neutralising H3N2 antibodies, these vaccines will not prevent population spread of a pandemic H3N2 variant. However, in the absence of prior neutralising antibody immunity, such as against an H7N9 virus, these vaccines may work very effectively within an individual which may impact population spread. Using the minimal model described, it was predicted that T cell-inducing vaccines would reduce transmission of novel influenza subtype such as H7N9 [42]. Therefore, T cell-activating vaccines may work best in the absence of prior antibody immunity to novel strains.

Current seasonal IIV vaccination strategies may even be hindering T cell immunity to IAV. In a mouse challenge model, it was observed that H3N2 vaccination led to more severe outcomes after H5N1 challenge and a reduction of recall of virus-specific CD8^+^ T cells at challenge [43]. Similarly, a comparison between repeatedly seasonally vaccinated children with cystic fibrosis and unvaccinated children showed that repeated IIV vaccination hindered the expansion of influenza-specific CD8^+^ T cell populations, however CD4^+^ T cell populations were unaffected [44]. This loss of natural immunity may hamper future IAV responses as previous studies have shown that natural immunity in children is broadly reactive and cross-reactive responses against H5N1 were observed in unexposed individuals [45]. While IIV for cystic fibrosis children is imperative for their protection against seasonal influenza, current vaccination strategies are failing to fully utilise the entire immunological armoury against IAV.

A recently identified CD8^+^ T cell epitope, PB1_413–421_, is restricted to highly prevalent HLA types (HLA-A*01:01, HLA-A*02:01, and HLA-B*37:01) and is >99.9% conserved across influenza A, B, and C viruses [46]. This could provide a priming target which may be protective in up to 54% of the global population based on HLA-coverage, illustrating the wide population potential of T cell activating vaccines [46]. This concept has been further explored in a ‘proof-of-principle’ study by Eickhoff et al. in which conserved HLA-supertype restricted IAV peptides were first identified by immune informatics [47]. Using this method, the authors identified 25 conserved MHC Class I peptide regions which were potential HLA-A2 restricted epitopes. Eickhoff et al. demonstrated that this peptide pool could stimulate IFNɣ responses in cultured PBMCs isolated from HLA-A2^+^ individuals and HLA-A2 transgenic mice were protected against lethal H1N1 challenge after vaccination with the peptide-DNA vaccine construct. However, the A2-restricted peptide pool did not stimulate responses in HLA-A2 negative individuals, and thus may not prime responses in other HLA types. Peptide-based T cell-activating vaccines are limited by common HLA types and due to bias in bioinformatic tools for peptide prediction there is a concern that individuals with rarer HLA types, such as those from ethnic minorities, would be excluded and responses could be subdominant or absent [48].

While the potential of T cell-activating vaccines is extremely promising, the effect of T cell-activating vaccine immune pressure on the IAV genome is yet to be fully understood. While some modelling studies suggest that the impact of T cell-activating vaccines on antigenic drift may be minimal [42], other studies have shown that selection by CD8^+^ T cells leads to increased mutations within epitope regions [49]. As such, interactions between vaccine enhanced immune pressure and the influenza genome still need to be further characterised but recognition and discussion of potential outcomes in the context of next generation vaccines is prudent.

Recognition of viral peptides requires presentation via MHC-peptide complexes, which necessitates cell entry for immune priming and activation and thus enables the potential for viral adaptation. As sterilising immunity is unlikely to be achieved by T cell-activating vaccines, the potential risk of immune evasion within an individual, and thus the population, must be seriously considered. Viral escape mutations in response to T cell pressure was previously observed in viral pathogens which cause persistent infection, such as hepatitis C virus (HCV) [50] and human immunodeficiency virus (HIV) [51,52]. Circumvention of T cell responses also has the potential to impede the responses of the other arms of the immune response. A previous study by Ciurea et al. also observed viral escape during lymphocytic choriomeningitis virus (LCMV) infection from neutralising antibody responses [53], which was determined to be a result of poor CD4^+^ T cell responses. This would indicate that the risk of potential T cell evasion is a credible concern and should be explored further.

Amino acid changes within IAV epitope regions have the potential to inhibit MHC-peptide loading into the peptide binding groove and obstructing presentation to T cells [54]. Additionally, mutations within T cell receptor (TCR) contact sites can affect T cell recognition of IAV peptides [55]. As universal T cell-activating vaccines can incorporate conserved internal IAV genes, which are not usually subjected to strong immune pressure during natural infection, this may lead to an increased likelihood of the emergence of variants. Previous studies which characterised an IAV NP escape mutant of the CD8^+^ T cell responses, D^b^NP_366_-N5H, arose naturally as early as day five of infection [56]. Infection with this NPN5H mutant virus reduced IAV-specific IFNɣ^+^ CD8^+^ T cell responses compared to the WT virus due to lower stability in the MHC cleft for presentation. Importantly, this mutant reverted to WT in the absence of T cell-mediated immune pressure, under MHC mismatch conditions. This study clearly demonstrates the capacity of traditionally conserved T cell epitope regions to subvert immune responses, which could be exacerbated by T cell-activating vaccination.

The avidity of T cell responses has also been shown to affect immune pressure towards conserved epitope regions. A CD8^+^ T cell epitope NP_418–426_, which is recognised by multiple HLA types [57], is considered to be a hypervariable region, with 4 identified mutable positions and over 20 different variants [58]. Three key mutants are associated with immune evasion, each requiring a separate pool of primed T cells with distinct T cell receptor repertoires [59,60] with different functional avidity [58]. As such, it is imperative that next generation vaccine design ensures that the targeting of epitope regions does not inadvertently facilitate the accumulation of amino acid substitutions which could lead to circumvention of the vaccinated T cell response.

T cell-activating vaccines are progressing through phase II clinical trials and are becoming a clinical reality in the near future. These include: Modified Vaccinia Ankara (MVA) vector incorporated with the full-length gene of NP and M1, the MVA-NP+M1 vaccine [61], and conserved multi-epitope peptide vaccine, FLU-v. The MVA-NP+M1 vaccine increases influenza specific T cell responses and can prevent infection or ameliorate disease severity in phase IIa clinical trials by challenge studies (n = 22) [62]. Antrobus et al. determined that MVA-NP+M1 had equivalent immunogenicity in adults (aged 18–45) and older adults (aged 50–85) [63], which may suggest this universal vaccine candidate can overcome the imprinting biases in adults, possibly due to a lack of HA content. While seroprotection rates after vaccination typically falls in adults over the age of 65 [64], the MVA-NP+M1 T cell-inducing vaccine induced robust CD8^+^ T cells against heterologous IAV proteins [63]. When used in conjunction with trivalent-inactivated influenza vaccine (TIV) the same authors found that together these vaccines could better boost antibody titres and influenza specific T-cell responses than when used alone [65]. Therefore, the MVA-NP+M1 CTL-inducing vaccine may have the capacity to build on prior immunity. Combination prime boost strategies with these vaccines are currently in phase IIb trials to explore these strategies [66]. However, the primary phase IIb trial of MVA-NP+M1 was stopped short due to a lack of reduced infection [67].

Recent trials of MVA-NP+M1 also aimed to assess its suitability as an intranasal influenza vaccine. In vitro assays using tonsillar mononuclear cells, isolated from both children and adults, showed that MVA-NP+M1 induced M1 protein expression in both B cells and tonsillar epithelial cells for local antigen presentation [68]. Characterisation of M1-specific CD8^+^ T cell responses using A2-M1_58–66_ tetramer staining demonstrated that MVA-NP+M1 significantly expanded CD8^+^ T-resident memory cell (T_RM_) populations, which can provide effective locally protective responses to IAV infection [69]. The peptide-based vaccine in development in collaboration with the NIH, named FLU-v, has been shown to elicit broad protection against influenza A and B viruses in phase IIb clinical trials by challenge studies (n = 153) [70]. The FLU-v vaccine contains four peptides derived from M1, NP from influenza A and B viruses, and M2 proteins with an oil in water adjuvant, Montanide ISA-51. A single dose of FLU-v significantly reduced the likelihood of H1N1pdm challenged individuals developing mild to moderate symptoms but did not significantly reduce viral shedding [70], which is consistent with other correlates of protection studies [71]. Thus, T cell-based vaccines may be in wider use soon given the rapid and recent progress of clinical trials in this area. However, as human challenge studies for both FLU-V [70] and MVA-NP+M1 [62] have shown, viral shedding in vaccinated challenge studies, a characterisation of breakthrough infections with full virus genome sequencing will be needed to assess the potential for viral evasion and adaptation to vaccine epitopes.

## 4. Adenovirus Vectors as Next Generation Vaccine Vectors

The use of replication defective adenovirus vectors (Ad) in next generation vaccines is promising for the induction of strong CTL and mucosal immune responses [72]. Adenovirus-vector based vaccines may be a more immunogenic alternative for immunising the immune to boost T cell responses. Adenoviral vectors are an attractive vaccine vector as the platform for non-replicating gene-based vaccines [73]. This family of DNA viruses naturally infects mucosal tissues, where immune recognition stimulates robust mucosal responses and further drives production of B cell and T cell responses [74]. Furthermore, they can infect a wide range of cell types, large DNA inserts can be incorporated with relative efficiency [75] and antigens can be processed for rapid direct- or cross-presentation via MHC I and II to CD8^+^ and CD4^+^ T cells [74]. Further modifications can be made to the vector to increase MHC antigen presentation and T cell priming such as cytokine integration [76].

Research is currently ongoing for the use of Ad vectors for immunisation against HIV-1 [77], Ebola [78], malaria [79], tuberculosis [80], SARS-CoV-2 [81], as well as influenza amongst others. If used widely, this exposure will lead to immune recognition of the vectors which could generate anti-vector neutralising antibodies, T cells (due to sequence homology particularly of the major surface hexon protein [82,83]) and type I IFN activated NK cells [84]. In the STEP HIV-1 vaccine trial, those with pre-existing Ad5 antibody titres were more likely to become infected with HIV-1 than those receiving placebo, possibly by an antibody dependent enhancement (ADE) type mechanism [85]. SARS-CoV-2 vaccines have also used Ad5 and Ad26 as viral vectors [86]. However, viral vectored vaccines can be impacted by prior immunity, with up to 85% of humans possessing pre-existing Ad5 specific antibodies [87], and widespread use will generate further anti-vector immunity [88]. However, over 100 different Ad vectors have been identified [89], thus alternatives can be used in the event of anti-vector antibody immunity impacting vaccine efficacy and should be further pursued in vaccine development. Furthermore, the route of vaccination may be used to circumvent anti-vector responses. For example, intranasal or oral Ad5 vaccination can generate cross-protective neutralising antibodies and T cell responses by circumventing pre-existing Ad5 immunity compared to parenteral vaccination [90]. Therefore, prime boost vaccination with alternating Ad vectors or routes of administration may be needed for effective T cell responses long term.

To further protect against the possibility of pre-existing immunity, non-human Ad could be used. Chimpanzee Ad from Oxford (ChAdOx1) has been used for rapid generation of a SARS-CoV-2 vaccine, with 70.4% vaccine efficacy in adults [81]. However, in a subset study, different efficacy rates were seen in high- versus low-dose groups, suggesting vector immunity in the high-dose group may limit the boosting effect to augment vaccine protection. The same vector ChAdOx1 expressing influenza NP and M1 proteins boosted T cell responses in human clinical trials [91] and can be used in combination with MVA-NP+M1. As a single dose of ChAdOx1 NP+M1 is not enough to maintain T cell responses long term in the elderly [91], a mixed vaccination model of MVA/ChAdOx1 NP+M1 was safe in human trials and elicited durable immune responses [91]. However, interim analysis of the phase IIb field trial of MVA+NP+M1 [67], after the receipt of IIV, did not improve protection rates. The trial was stopped before the second year as it was unlikely to meet primary endpoints and achieve the required level of reduction in the incidence of laboratory-confirmed influenza compared to standard vaccination. Therefore, viral vectored universal vaccines are back to the drawing board for the route, combination, and design needed to improve protection above the standard of care for seasonal influenza.

## 5. HA-Stem Vaccine and Immune Pressure

The haemagglutinin (HA) stem region is an attractive target for next generation vaccine development as this has functional constraints for adaptation and remains relatively conserved [92]. Functional conservation enables recognition by broadly neutralising antibodies (bnAbs) across multiple IAV strains [93] and can confer protection across multiple influenza subtypes [94]. A universal vaccine which primes against conserved stalk regions has recently published phase I clinical trial results with robust immunogenicity as a chimeric live attenuated vaccine and inactivated vaccine with AS03 adjuvant [95]. This chimeric HA-based vaccine elegantly combines varying HA head regions to a H1 stem in a repeated prime/boost regimen to stimulate conserved immune responses to the immune-subdominant stem region. After testing multiple vaccination combinations, it was evident that priming with an influenza B virus presenting chimeric H9 followed by H8/1 LAIV and H5/1 IIV resulted in the highest levels of protection against H1N1 challenge in mice [95]. However, whilst immunogenicity was evident from these vaccine approaches in phase 1 clinical trials and animal models, the conservative levels of antibody boosting in phase 1 interim analysis may not yield improved supra-seasonal protection and GlaxoSmithKline has suspended further clinical trials with the chimeric HA-adjuvant vaccine approach [96]. Alternative strategies to elicit HA stalk responses should be pursued with lessons learnt from previous trials.

While the HA stem region is more conserved in comparison to the globular head region, mutations can arise in the presence of bnAbs leading to reduced antibody recognition [97]. Therefore, increased pressure directed towards the HA stem could lead to a loss of antibody recognition of new variants across a vaccinated population. A recent study by Park et al. found that the HA polymorphism A388V, which arose naturally during cell culture, was being selected for during infection in individuals with higher pre-existing anti-HA stem IgG [98]. Further investigation found that serum from the study participants had reduced recognition of the mutant stem compared to wildtype (WT) and modelling suggested that while the HA-A388V mutation did not directly interact with bnAbs, it affected the structure of the α-helix of the stem region [98]. When WT and the HA-A388V mutant strain were co-cultured together at a 50:50 ratio in the presence of bnAb CR6261, the mutant strain out competed the WT strain within 72 h to become the dominant virus [98]. These findings demonstrate that mutants within the HA stem region can arise as a response to immune pressure against conserved regions. As pre-existing HA stem antibodies correlate with mutant virus selection in this study, it suggests that enhancing immune pressure by priming against the HA stalk region may risk further driving immune evasion. Therefore, HA stalk-based vaccines may need to utilise mosaic approaches in order to circumvent virus escape.

The development of mosaic vaccines [99,100] may become increasingly important as a way of creating broader protection against divergent strains to pre-empt escape and drive broadly reactive responses. Mosaic vaccines prime for multiple versions of the same IAV antigenic target across differing strains and phylogenic groups. This strategy may also reduce the generation of non-neutralising antibodies by priming for multiple IAV subtypes and diminish the chance of developing enhanced respiratory disease (ERD). While a mosaic T cell-activating nanoparticle type vaccine is not currently in development, a chimeric HA stem vaccine has already been shown to produce strong group 1 heterosubtypic immunity [95], and mosaic HA nanoparticles are in clinical trials [100]. Novel mRNA vaccine technology which has been very successful for COVID-19 is planned for universal influenza vaccine development [101], is also amenable to mosaic design, and can prime T cell responses due to protein translation at vaccination.

The COVID-19 pandemic has demonstrated the high efficacy of mRNA-based vaccines compared to other vaccine formats. mRNA-based influenza vaccines have also long been in development and may be key to universal vaccine design [102]. mRNA vaccines have an advantage of being self-adjuvating for TLR 3, 7, and 8, whilst incorporating nucleoside modifications [101] avoids excess inflammation through the TLR system. This also increases protein production [103] and the mRNA lipid nanoparticle vaccine formulation improves delivery and stability [104,105]. Vaccine formulations including HA of pandemic potential [104], and combination strategies of HA, NA, NP, [106] and M2e [101], have shown increased protection in comparison to standard of care inactivated vaccines across multiple species [104,106]. Human trials include HA pandemic potential IAV subtypes [107] and seasonal QIV HA-based formulations [108]. The combination of influenza QIV and COVID-19 vaccines by Moderna [108] in an ongoing COVID-19 vaccinated phase I/II trial will establish the ability to boost prior immunity with mRNA vaccines. Whilst prime-boost dosing appears to increase immunogenicity [101,106], dose sparing may be possible for doses [109]. Whilst the mRNA strategy is clearly immunogenic, HA only platforms will only address single pandemic potential viruses like recombinant HA vaccines, thus employing multivalent antigen combination approaches or mosaic HA within mRNA vaccines holds the most promise.

## 6. NA as a Next Generation Vaccine Target

Neuraminidase (NA) has also been proposed as a potential target for next generation IAV vaccines (reviewed in [110]). Presented on the surface of IAV, alongside HA, NA is prone to seasonal antigenic drift, but this occurs independently of HA mutations [111]. NA-inhibiting antibodies are also an independent correlate of HA-inhibiting antibodies for vaccine mediated protection in LAIV and IIV [112]. Protection against novel pandemic strains by recognition of a previously circulating NA subtype has been observed previously. In 1969 Schulman et al. demonstrated that previous exposure to H2N2 generated protection against H3N2 challenge in mice [113], leading to the proposal that the IAV taxonomy needed to consider both HA and NA lineages.

Currently, NA content is not regulated in vaccine content, and protective titres for NA antibodies are yet to be clearly defined. Studies using experimental NA-only vaccines have shown that these vaccines can be successful in generating immunity against IAV. A human challenge study in 1974 of N2-specific vaccinated participants had reduced viral titres in nasal washes after H3N2 infection but did not prevent infection during the initial challenge [114]. More recent H5N1 challenge studies found that using a H1N1-NA DNA vaccine offered partial protection [45] and intranasal inoculation with an N1 virus-like particle (VLP) protected against lethal challenge [115]. However, Eichelberger and Monto in a 2019 review [116] noted that in unpublished studies some NA antibodies were seen to have poor inhibition and were not protective in immunocompromised mice. This suggests that NA antibodies need to be further characterised and may not be protective in vulnerable populations.

As NA facilitates viral release through the cleavage of HA, targeting of NA by next generation vaccines would still allow HA-mediated viral entry into host cells and potential adaptation of IAV. This could lead to mutations conferring resistance against clinically important antiviral drugs such as oseltamivir, a neuraminidase inhibitor (NAI). Conformational changes in the NA protein can prevent the binding of NAIs and reduce efficacy. NAIs are a common clinically available antiviral drugs to treat influenza and antiviral resistance has already been observed in circulating IAV [117,118,119]. Some studies suggest that there has been limited transmission of antiviral resistant IAV within the community to date, with only 1% of H1N1pdm strains containing known drug resistance mutations during the 2010/11 season [120]. Most IAV sequences with antiviral resistance mutation NA-H275Y identified in this study were from individuals who had been treated with oseltamivir prior to sample collection. As such, it is important to investigate if enhanced NA-directed immune pressure across a population alongside oseltamivir usage could drive further IAV adaptation within the community.

Another important consideration would be if NA-directed immune pressure could lead to compensatory mechanisms in other IAV genes. A study by Ilyushina et al. characterised previously identified mutations which arose within HA as a response to NAI use and observed that these lead to diminished antibody responses [121]. H1N1 viruses with the HA-G155E and HA-D222G mutations could replicate to significantly higher viral titres in the presence of oseltamivir and had a 20-fold reduction in antibody reactivity [121]. These findings suggest that directed pressure towards NA can lead to subversion of pre-existing immunity, which must be further explored when developing next generation vaccines.

Directed pressure towards HA may also similarly affect NA. Recent studies have seen that bnAbs targeting HA stem regions can inhibit NA function due to steric interference [122,123]. This contribution of NA inhibition aiding the protection elicited by HA stem bnAbs could provide a two-layer strategy of combining both HA and NA as targets in next generation vaccines for synergistic effects. However, the ‘push and pull’ relationship between HA and NA in terms of mutations and compensatory mechanisms still needs to be further understood.

## 7. Foreseeing Unexpected Outcomes of Universal Vaccination

It is important to prepare for unexpected outcomes of universal vaccination ahead of time and vaccine enhanced respiratory disease (VAERD) could be a potential obstacle for some next generation vaccine strategies. Enhanced pneumonia and disease severity was observed in pigs that had been vaccinated with a H1N2 vaccine and subsequently challenged with H1N1 [124]. Further investigation determined this was caused by vaccine induced anti-HA2 antibodies [125], which could indicate this could be a hurdle for bnAb vaccines designed to target HA stem regions. It has also been seen that the presence of non-neutralising antibodies can be detrimental during influenza infection [126] and leading to ERD. This suggests that ADE could be an underappreciated factor of IAV infection. It has been previously theorised that sub-neutralising antibody levels and activation of FcγRI and FcγRIIA could promote viral cell entry, replication, and subsequent antigen presentation and immunogenicity of live attenuated viral vaccines [127]. However, a recent review has noted that in the context of influenza, LAIVs have been repeatedly shown to be safe and effective in animal models with no indications of VAERD or ADE [128]. A study by Winarski et al. sought to evaluate the effect of monoclonal antibodies (mAbs) on influenza disease in a mouse model [129]. They pre-treated mice with mAbs specific for the globular head regions of H3N2 (mAbs termed: 78/2, 69/1) and subsequently challenged mice with a non-lethal H3N2 challenge. Treatment with mAbs was seen to have a detrimental effect on lung pathology and mAb 78/2 increased lung viral titre at some treatment doses. Virus fusion kinetics was promoted in MDCK cell culture after mAbs treatment and could be the mechanism of their ADE effect for ERD. The potential for ADE by next generation vaccines has been assessed in preclinical studies with no adverse effects so far [130].

The presence of non-neutralising antibodies could have further consequences outside of ADE that are still less understood. A previous study by Wanzeck et al. identified that mice first challenged with a highly glycosylated HA IAV variant and then subsequently challenged with WT virus suffered increased weight loss and lung immunopathology, which was abated in T cell-depleted conditions [131]. This effect was also observed in mice primed with seasonal H1N1 and given a secondary challenge of H1N1pdm, which could indicate that pre-existing T cell immunity may also play a factor in unexpected universal vaccination outcomes. Wanzeck et al. proposed that this was induced by a mismatch between neutralising antibody and T cell responses, the mechanism of which still needs to be defined. It could be important to further define non-neutralising antibody and T cell dynamics in the context of next generation vaccines which enhance T cell responses, as cross-reactive memory specific T cells have been shown to induce acute lung injury during IAV infection [132], and to cause bystander damage to uninfected cells in vitro [133]. However, a relationship between vaccine-induced T cells and VAERD has not been well defined by local or peripheral vaccination. The induction of local T cell resident memory (TRM) requires local antigen presentation within the lung and nasal passages [134] to seed local TRM, with protection expiring 7 months after infection [135] and long-term antigen persistence from DNA virus vectored vaccines extending TRM populations [136]. So far in animal models, vaccine-induced T cells have been essential for protection rather than implicated in VAERD, but further research is needed given the observations of Wanzeck et al. about HA mismatch and the potential for uncontrolled IFNγ production by T cells, especially by TRM populations within the lung.

Unfortunately, phase II trials of viral vectored T cell-activating and phase I trials of HA chimeric-stem vaccines have stalled recently due to a lack of improved efficacy or substantially elevated immunity. Other strategies such as FLU-v and Ferritin HA nanoparticle remain in progress for clinical trials, amongst over 2000 clinical trials for influenza vaccines and drugs. A vaccine which can ultimately combine broadly reactive antibodies and T cells for conserved epitopes to provide two-layer protection from viral entry, or immune recognition of virus infected cells would be ideal and as evident from the recent futility of trials of viral vectored T cell-activating and HA chimeric-stem vaccines, their use alone may not be enough to improve the standard of care from inactivated vaccines. Combined approaches may further augment responses with MVA-NP+M1 with chimeric HA in pre-clinical animal models [137], and should be explored with further vaccine approaches to broaden anti-influenza immunity.

## 8. Conclusions: Ways Forward for Universal Vaccines

The road to universally ‘immunising the immune’ with next generation vaccines is complex and must build on prior immunity, but important steps and discoveries are already underway. An overview of potential conditions outlined within this review that next generation universal vaccines must overcome are highlighted in Figure 1 and summarised in Table 1. Different vaccine platforms will have different hurdles in relation to both pre-existing immunity and viral strain challenge. Individuals vaccinated with a broadly neutralising HA stem vaccine (Figure 1A,B) may have different levels of efficacy depending on their prior challenge history. Immune imprinting and subsequent challenge with a similar strain post vaccination would lead to enhanced immunity but prior imprinting and heterologous challenge can impede immune responses [26,27]. Pre-existing immunity in individuals vaccinated by T cell-activating vaccines (Figure 1C,D) may also have different outcomes, with previous studies showing pre-existing antibodies may suppress cross-reactive T cell responses [37] and pre-existing immunity to one IAV subtype could affect responses to novel variants of the same subtype [42]. Cross-reactive T cell responses may also lead to unexpected side effects after heterosubtypic challenge, which requires further investigation and characterisation [129,131]. Additionally, some populations with rare HLA types would not be optimally protected by the targeting of unrelated HLA supertypes [47,48]. To circumvent these issues, mosaic-based vaccine design may be able to account for prior immune history. Prior exposure to adenovirus vaccine vectors will also impact on vaccine efficacy (Figure 1E,F) and repeated vaccination may increase vector immunity. This pre-exposure would impair priming against the targeted antigen as responses would more heavily target the vector portion of the vaccine, leading to reduced immunity and a failure to generate sufficient memory responses [85,87]. There is no doubt that universal IAV vaccines are needed to improve global preparedness for not only seasonal but also pandemic influenza strains. However, it is vital that next generation vaccines protect against novel variants and not drive their selection for vaccine immune evasion due to residual virus replication under intense immune pressure. Enhanced influenza surveillance of all 8 genes will be paramount for the rapid detection of emerging variants that could circumvent vaccine-mediated protection.

Successful universal vaccination across a population may require a multi-layer approach of simultaneously utilising bnAbs targeting the HA head and stem region, surface NA and eliciting strong T cell responses as a combined vaccine platform. By inducing cross-protective antibody responses to both surface antigens and recognition of conserved internal peptides by T cells, potential routes for breakthrough variants could be reduced. By using multiple immunising strategies in concert with each other, a multi-faceted response may protect a larger proportion of an immunologically varied population. Employing multiple arms of the immune response may help to ‘future-proof’ vaccines against novel variants [138]. Next generation universal influenza vaccines are fast becoming a reality and have the capacity to have a huge beneficial impact on human health, but as influenza viruses readily adapt, our strategies must be variant proof.

## Figures and Tables

**Figure 1 viruses-13-01779-f001:**
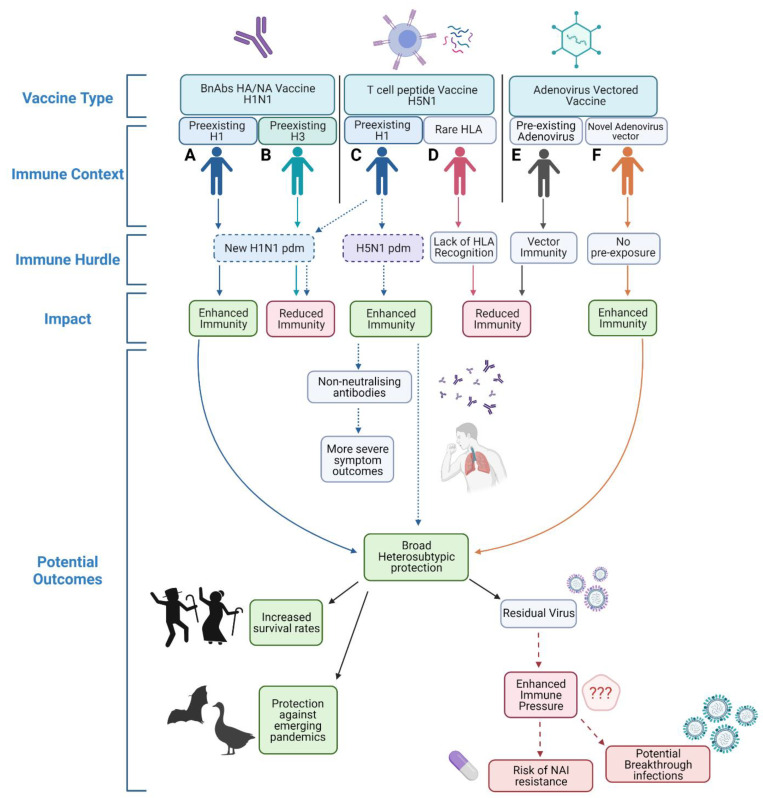
*Overview of potential hurdles to varying universal IAV vaccination strategies.* Individual (**A**) vaccinated with a H1N1 BnAbs HA/NA vaccine with prior H1 exposure would have enhanced immunity when exposed to a novel H1N1 pdm virus. Conversely, individual (**B**) also vaccinated with a H1N1 BnAbs HA/NA vaccine but with prior H3 exposure may have reduced immunity against the same novel H1N1 pdm virus. After vaccination with a H5N1 T cell peptide vaccine individual with prior H1 immunity (**C**) would have enhanced protection against a homosubtypic H5N1 challenge but may have reduced protection against a novel H1N1 pdm virus. Individual (**D**) who possesses a rarer HLA type may fail to be protected by a T cell peptide vaccine due to lack of T cell recognition. After vaccination with an adenovirus vectored vaccine an individual with prior vector immunity (**E**) would have impartial priming leading to reduced immunity, whereas an individual with no prior adenovirus vector exposure (**F**) would be sufficiently primed. Created in biorender.com.

**Table 1 viruses-13-01779-t001:** Summary of potential outcomes and hurdles for next generation influenza vaccines.

Vaccine Type	Example Vaccine	Baseline Immunity	Hypothetical Hurdle	Impact	Potential Outcomes	References
bnAbs HA/NA vaccine (H1N1)	Chimeric HA LAIV + AS03 (Phase I trial halted)	H1N1 primed	New H1N1 pdm virus	Enhanced immunity	Broad heterosubtypic protection	[93,94,95]
		H3N2 primed		Reduced immunity	Impartial protection	[26,27]
T cell peptide vaccine (H5N1)	FLU-V (Phase IIb trial ongoing)	H1N1 primed		Reduced immunity	Impartial protection	[37,42]
		H1N1 primed	H5N1 pdm virus	Enhanced immunity	Broad heterosubtypic protection	[62,70]
		Rare HLA-type	Lack of T cell priming	Reduced immunity	Impartial protection	[47,48]
Adenovirus vectored vaccine	ChAdOx1 NP + M1 (Phase I trial completed)	Preexposure to vector adenovirus	Vector Immunity disrupting priming	Reduced Immunity	Impartial protection	[85,87]
		Naïve to adenovirus vector	No pre-exposure	Enhanced Immunity	Broad heterosubtypic protection	[72,74,91]

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
