# Peer review of "Universally Immune: How Infection Permissive Next Generation Influenza Vaccines May Affect Population Immunity and Viral Spread"

_viruses, 2021, doi:10.3390/v13091779_

Round 1

Reviewer 1 Report

Since the 2009 H1N1 pandemic, there has been renewed interest in the development of a universal influenza virus vaccine to overcome the limitation of current seasonal vaccines. As a result, there are myriad of vaccines candidates in clinical development that leverage epitopes and platforms fundamentally different from inactivated vaccines in eggs. The review by Bull et al. explores select universal HA-stem, adenoviral, and T-cell strategies and examines their potential strengths and weaknesses with a particular emphasis on how candidate might influence immune escape. Overall, the piece is a timely contribution to the literation and would be of broad interest to the readership of Viruses. The following revisions are warranted before the article be accepted for publication.

Major concerns:

  • The title of the manuscript should be modified to better reflect the discussion of the vaccine platforms. More specifically the use of the term “non-neutralizing” is problematic as many antibodies which target the HA stalk domain are in fact neutralizing. It would seem as the intention is distinguish between vaccine induce immunity that does not block viral entry (i.e. HAI+ antibodies or T-cell responses) but again as the name suggests broadly neutralizing antibodies are in fact neutralizing!

  • The discussion of MVA-NP+M1 (Section 4) and chimeric HAs (Section 5) needs to be updated to better reflect the current progress of clinical trials with these candidates. For example, interim analysis of the Phase IIb field trial of MVA+NP+M1 (https://clinicaltrials.gov/ct2/show/NCT03880474) determined that the trial was unlikely to meet primary endpoint and achieve the required level of reduction in the incidence of laboratory-confirmed influenza. Therefore, the trial was not recommended to continue for a second season. Similarly, GSK is no longer pursuing the chimeric hemagglutinin strategy (https://www.fiercebiotech.com/biotech/gsk-dumps-universal-flu-vaccine-after-interim-data-readout).

  • Section 6 should be expanded to include a discussion of concerns surrounding T-cell vaccines related to immunopathology.

Minor comments:

  • Line 20: diverse is used twice in the same sentence, revise
  • Line 29: spell out IAV before abbreviating
  • Line 44: This is (a) respite?
  • Line 56: A more fulsome explanation of what is meant by the term heterosubtypic immunity is warrant. To this end It could be helpful to include a few sentences of background on the number of subtypes of IAVs in the first paragraph of the introduction.
  • Line 82: The topic of OAS has been reviewed in great detail, additional references should be included (https://pubmed.ncbi.nlm.nih.gov/31964645/, https://pubmed.ncbi.nlm.nih.gov/30617114/, https://pubmed.ncbi.nlm.nih.gov/28398521/)
  • Line 110: More details for this study need to be included for proper context: better protection than what? IIV? Natural infection?
  • Section 3: There is ample evidence in mice that heterosubtypic protection is mediated by CD8+ T cell and increasing evidence that elevated T-cells in humans can play a role in control infection. A few sentences summarizing these data would improve this
  • Sentence from Line 134-136 needs revision
  • Line 147: “they” is reads informally, consider using “the authors”
  • Line 180: write T-cell receptor out in full before abbreviation
  • In vitro and et al. should be italicized
  • More information on FLU-v is needed; description of antigens in this vaccine candidate is lacking in the text
  • Line 235: subject verb agreement
  • Line 241: this sentence should be moved to the beginning of the paragraph to better explain the adenoviral serotypes
  • Section 4: Section title reads, “Adenoviral virus next generation vaccine vectors” yet only adenoviral vectors are discussed.
  • Line 276: Subject verb agreement
  • Paragraph that starts one Line 295 should be more balanced to reflect data which indicates that there are inherent constraints on the ability of HA2 to mutate (https://pubmed.ncbi.nlm.nih.gov/24277853/)
  • Sentence beginning on line 308 needs revision
  • Line 318: ERD needs to be written out in full before abbreviation
  • Line 335: ADE needs to be written out in full before abbreviation

Author Response

Reviewer 1:

Since the 2009 H1N1 pandemic, there has been renewed interest in the development of a universal influenza virus vaccine to overcome the limitation of current seasonal vaccines. As a result, there are myriad of vaccines candidates in clinical development that leverage epitopes and platforms fundamentally different from inactivated vaccines in eggs. The review by Bull et al. explores select universal HA-stem, adenoviral, and T-cell strategies and examines their potential strengths and weaknesses with a particular emphasis on how candidate might influence immune escape. Overall, the piece is a timely contribution to the literation and would be of broad interest to the readership of Viruses. The following revisions are warranted before the article be accepted for publication.

Major concerns:

  • The title of the manuscript should be modified to better reflect the discussion of the vaccine platforms. More specifically the use of the term “non-neutralizing” is problematic as many antibodies which target the HA stalk domain are in fact neutralizing. It would seem as the intention is distinguish between vaccine induce immunity that does not block viral entry (i.e. HAI+ antibodies or T-cell responses) but again as the name suggests broadly neutralizing antibodies are in fact neutralizing!
  1. We thank the reviewer for this important point, and have updated out title that these vaccine strategies are infection permissive, as they do not block virus entry. HA stalk antibodies typically require much higher orders of magnitude concentrations for equivalent virus neutralisation.

From: Universally immune: how will non-neutralising next generation influenza vaccines affect population immunity and viral spread

To: Universally immune: how infection permissive next generation influenza vaccines may affect population immunity and viral spread

  • The discussion of MVA-NP+M1 (Section 4) and chimeric HAs (Section 5) needs to be updated to better reflect the current progress of clinical trials with these candidates. For example, interim analysis of the Phase IIb field trial of MVA+NP+M1 (https://clinicaltrials.gov/ct2/show/NCT03880474) determined that the trial was unlikely to meet primary endpoint and achieve the required level of reduction in the incidence of laboratory-confirmed influenza. Therefore, the trial was not recommended to continue for a second season. Similarly, GSK is no longer pursuing the chimeric hemagglutinin strategy (https://www.fiercebiotech.com/biotech/gsk-dumps-universal-flu-vaccine-after-interim-data-readout).
  1. We thank the reviewer for this important update on the status of these clinical trials and have updated our text accordingly.

Section 4: final paragraph:

“However, interim analysis of the phase IIb field trial of MVA+NP+M1 [67] after the receipt of IIV, did not improve protection rates. The trial was stopped before the second year as it was unlikely to meet primary endpoints and achieve the required level of reduction in the incidence of laboratory-confirmed influenza compared to standard vaccination. Therefore, viral vectored universal vaccines are back to the drawing board for the route, combination and design needed to improve protection above the standard of care for seasonal influenza.”

Section 5:

“However, whilst immunogenicity was evident from these vaccine approaches in phase 1 clinical trials and animal models, the conservative levels of antibody boosting in phase 1 interim analysis may not yield improved supra-seasonal protection and GlaxoSmithKline has suspended further clinical trials with the chimeric HA adjuvant vaccine approach [96]. Alternative strategies to elicit HA-stalk responses should be pursued with lessons learnt from previous trials.”

  • Section 6 should be expanded to include a discussion of concerns surrounding T-cell vaccines related to immunopathology.
  1. We appreciate the reviewer comment, and in section 6, T cell immunopathology and HA imbalance is discussed. Further discussion of bystander activation and damage has been added and T cell resident memory by intranasal vaccines.

“This effect was also observed in mice primed with seasonal H1N1 and given a secondary challenge of H1N1pdm, which could indicate that pre-existing T cell immunity may also play a factor in unexpected universal vaccination outcomes. Wanzeck et al. proposed that this was induced by a mismatch between neutralising antibody and T cell responses, the mechanism of which still needs to be defined. It could be important to further elucidate these non-neutralising antibody and T cell dynamics further for next generation vaccines which enhance T cell responses as cross-reactive memory specific T cells have been shown to induce acute lung injury during IAV infection [121], and to cause bystander damage to uninfected cells in vitro [122]. However, a relationship between vaccine-induced T cells and VAERD has not been well defined by local or peripheral vaccination. The induction of local T cell resident memory (TRM) requires local antigen presentation within the lung and nasal passages [123] to seed local TRM, with protection expiring 7 months after infection [124] and long-term antigen persistence from DNA virus vectored vaccines extending TRM populations [125]. So far in animal models, vaccine induced T cells have been essential for protection rather than implicated in VAERD, but further research is needed given the observations of Wanzeck et al. about HA mismatch and the potential for uncontrolled IFNg production by T cells, especially by TRM populations within the lung.”

Minor comments:

  • Line 20: diverse is used twice in the same sentence, revise
  • Diverse -> plethora
  • Line 29: spell out IAV before abbreviating
  • Updated
  • Line 44: This is (a) respite?
  • Respite -> reprieve
  • Line 56: A more fulsome explanation of what is meant by the term heterosubtypic immunity is warrant. To this end It could be helpful to include a few sentences of background on the number of subtypes of IAVs in the first paragraph of the introduction.
  • Added:

“cross reactive heterosubtypic immunity for different influenza strains and subtypes for an extended duration. Whilst influenza A viruses have at least 17 different HA and 11 NA subtypes and viruses continually drift the coverage of the universal vaccine field varies by vaccine type. Heterosubtypic vaccines can be either pan-influenza covering both influenza A and B viruses, group-specific based on HA phylogeny (e.g. group 1 HA H1N1/H5N1 versus group 2 HA H3N2/H7N9), or subtype specific (different H1N1 strains) depends on the breadth of responses elicited and epitope conservation that is targeted by those vaccines.”

  • Line 82: The topic of OAS has been reviewed in great detail, additional references should be included (https://pubmed.ncbi.nlm.nih.gov/31964645/, https://pubmed.ncbi.nlm.nih.gov/30617114/, https://pubmed.ncbi.nlm.nih.gov/28398521/)
  • Added

  • Line 110: More details for this study need to be included for proper context: better protection than what? IIV? Natural infection?
  • Added: from acquisition of infection

  • Section 3: There is ample evidence in mice that heterosubtypic protection is mediated by CD8+ T cell and increasing evidence that elevated T-cells in humans can play a role in control infection. A few sentences summarizing these data would improve this
  • Added: “T cells have been found to have essential protective functions from severe disease [39], reducing viral loads [40] and reducing symptom duration and fever [41], especially in the absence of neutralizing antibodies where T cells exhibit remarkable cross-reactivity to novel strains.”

  • Sentence from Line 134-136 needs revision

Changed to “This loss of natural immunity may hamper future IAV responses as previous studies have shown that natural immunity in children is broadly reactive and cross-reactive responses against H5N1 has been observed in unexposed individuals [45].”

  • Line 147: “they” is reads informally, consider using “the authors”
  • Updated to the authors/ Eickhoff et al

  • Line 180: write T-cell receptor out in full before abbreviation
  • Added

  • In vitro and et al. should be italicized
  • Updated throughout, and de novo

  • More information on FLU-v is needed; description of antigens in this vaccine candidate is lacking in the text
  • Added: The FLU-V vaccine contains 4 peptides derived from M1, NP from influenza A and B viruses, and M2 proteins with an oil in water adjuvant: Montanide ISA-51.

  • Line 235: subject verb agreement

Amended

  • Line 241: this sentence should be moved to the beginning of the paragraph to better explain the adenoviral serotypes
  • Moved this section up as suggested

  • Section 4: Section title reads, “Adenoviral virus next generation vaccine vectors” yet only adenoviral vectors are discussed.
  • Updated to “Adenoviral vectors for vaccines”

  • Line 276: Subject verb agreement
  • Amended

  • Paragraph that starts one Line 295 should be more balanced to reflect data which indicates that there are inherent constraints on the ability of HA2 to mutate (https://pubmed.ncbi.nlm.nih.gov/24277853/)
  • Updated to: The haemagglutinin (HA) stem region is an attractive target for next generation vaccine development as this region has functional constraints for adaptation and remains relatively conserved, which can be recognised by broadly neutralising antibodies (bnAbs) across multiple IAV strains [84] and have been previously shown to confer protection across multiple influenza subtypes [85].

  • Sentence beginning on line 308 needs revision
  • Revised
  •  
  • Line 318: ERD needs to be written out in full before abbreviation
  • Updated

  • Line 335: ADE needs to be written out in full before abbreviation
  • Updated

Reviewer 2 Report

The review by Bull et al. (Universally immune: how will non-neutralising next generation influenza vaccines affect population immunity and viral spread) explores the information available about different next-generation influenza vaccines. An important amount of information has been compiled and major influenza alternatives vaccines platforms are covered. However, a more in detail analysis must be provided in some cases and it must be established very clear the current state of development in each strategy. My major and minor comments are listed below:

Major comments:

The manuscript gives the impression that next-generation vaccines focus only on non-neutralizing responses, whereas in some cases, there is a combination between neutralizing and non-neutralizing antibodies plus the activation of the cellular immune response. The fact that probably the best vaccine approach will be the one able to elicit neutralizing and non-neutralizing antibodies plus cellular immune activation is never stated in the manuscript. Additionally, it gives the impression that all strategies are at similar level of development, which is misleading considering that a LAIV platform vaccine is currently approved in the USA. Furthermore, a similar approach has been used for several years in Russia for instance. The section about LAIV currently approved, under evaluation and novel platforms should be discussed more in detail. More comparisons with IIV, the potential duality with humoral and cellular immune response and the mimicking of the infection are topics that should be covered. Although LAIV are not evaluated by the presence of neutralizing antibodies, they are also capable to generate this type of antibodies so the information about these types of vaccines should be discussed further. Novel LAIV platforms evaluated in animal models or under evaluation in humans should be mentioned.

In the LAIV section, the IgA portion should be discussed more in detail due to the higher cross-reactive observed with IgA and the relation with mucosal immunity. Additionally, the lower efficacy of LAIVs is explained by the authors just a consequence of the OAS but an explanation related to the type of LAIV approved is completely ignored. Could it be that the currently approved LAIV is over attenuated impacting the immunogenicity?

Authors mention that no increased rate of protection is observed between LAIV and IIV however, the reference cited stated clearly that no conclusions are possible due to the low incidence of influenza. Additionally, the study was conducted just in a population 60 and older.

Vaccines based on mRNA should be discussed more in detail. Several articles have described the potential use of mRNA vaccines for Influenza

Adenovirus section: little information about Influenza vaccines is presented and most of the information is related to SARS-CoV-2, which is not the topic of the review

HA-stem vaccine: What about ADE in the case of this vaccines? Authors mentioned that this type of vaccines have been evaluated in LAIV and IIV platforms. Any difference in the outcome

In general, more figures should be added to exemplify the information provided, which would make the review more valuable. At least, a table summarizing the different strategies discussed, pros and cons and state of evaluation in clinical trials (if it is possible should be provided). Information about the different platforms and animal models would be informative. Figure 1 is hard to understand, and no nomenclature is provided in the figure legend.

Why neuraminidase (NA)-based vaccines were not discussed?

Minor comments:

Line 32: Please remove ‘’…threat with higher mortality rate...’’

Line 33: replace term accrue

Line 44: Reference is missing

Line 116: Please replace utilization

Line244-245: References are missing

Line 327: It gives the impression that none of the vaccines discussed have reached approval which is not true. Please rephrase.

Line 286-294 This paragraph seems to be out of place. I would recommend reorganize the text

Author Response

Reviewer 2:

The review by Bull et al. (Universally immune: how will non-neutralising next generation influenza vaccines affect population immunity and viral spread) explores the information available about different next-generation influenza vaccines. An important amount of information has been compiled and major influenza alternatives vaccines platforms are covered. However, a more in detail analysis must be provided in some cases and it must be established very clear the current state of development in each strategy. My major and minor comments are listed below:

Major comments:

The manuscript gives the impression that next-generation vaccines focus only on non-neutralizing responses, whereas in some cases, there is a combination between neutralizing and non-neutralizing antibodies plus the activation of the cellular immune response. The fact that probably the best vaccine approach will be the one able to elicit neutralizing and non-neutralizing antibodies plus cellular immune activation is never stated in the manuscript.

  • Added to the discussion of the failed trials combined approaches and 2-layer immunity.

“Unfortunately, phase II trials of viral vectored T cell activating and phase I trials of HA-chimeric stem vaccines have stalled recently due to a lack of improved efficacy or substantially elevated immunity. Other strategies such as FLU-v and Ferritin HA nanoparticle remain in progress for clinical trials, amongst over 2,000 clinical trials for influenza vaccines and drugs. A vaccine which can ultimately combine broadly reactive antibodies and T cells for conserved epitopes to provide 2-layer protection from viral entry, or immune recognition of virus infected cells would be ideal, and as evident from the recent futility of trials viral vectored T cell activating and HA-chimeric stem vaccines, their use alone may not be enough to improve the standard of care from inactivated vaccines. Combined approaches may further augment responses with MVA-NP+M1 with chimeric HA in pre-clinical animal models [126], and should be explored with further vaccine approaches to broaden anti-influenza immunity.”

Additionally, it gives the impression that all strategies are at similar level of development, which is misleading considering that a LAIV platform vaccine is currently approved in the USA. Furthermore, a similar approach has been used for several years in Russia for instance. The section about LAIV currently approved, under evaluation and novel platforms should be discussed more in detail. More comparisons with IIV, the potential duality with humoral and cellular immune response and the mimicking of the infection are topics that should be covered. Although LAIV are not evaluated by the presence of neutralizing antibodies, they are also capable to generate this type of antibodies so the information about these types of vaccines should be discussed further. Novel LAIV platforms evaluated in animal models or under evaluation in humans should be mentioned.

  • LAIV is discussed in detail in page 3 line 103-129. The chimeric LAIV is discussed as part of Section 5, line 366-375.

In the LAIV section, the IgA portion should be discussed more in detail due to the higher cross-reactive observed with IgA and the relation with mucosal immunity. Additionally, the lower efficacy of LAIVs is explained by the authors just a consequence of the OAS but an explanation related to the type of LAIV approved is completely ignored. Could it be that the currently approved LAIV is over attenuated impacting the immunogenicity?

  • Added to section 2

Whilst original antigenic sin may account for this phenomenon, it is also possible that the available strains for LAIV are too attenuated for efficient viral replication in the upper nasal passages to prime cellular immunity and high titer mucosal antibodies.

Authors mention that no increased rate of protection is observed between LAIV and IIV however, the reference cited stated clearly that no conclusions are possible due to the low incidence of influenza. Additionally, the study was conducted just in a population 60 and older.

  • The important caveat to the study has been added.

“However, in a small study of Flumist in South Africa over 60-year-olds, there is no increased rate of protection from acquisition of infection or duration of immunity for LAIV compared to IIV, however low incidence of infection may preclude solid evidence [30]. “

Vaccines based on mRNA should be discussed more in detail. Several articles have described the potential use of mRNA vaccines for Influenza

  • The discussion of mRNA vaccines is included in page 7, line 415-418.

Adenovirus section: little information about Influenza vaccines is presented and most of the information is related to SARS-CoV-2, which is not the topic of the review.

  • The MVA influenza vaccine discussed throughout, and focus of section 3. Additional information has been added to the progress of influenza trials to the section 3 and 4. We hope this sufficiently covers influenza vaccination and topical COVID-19 vaccination.

HA-stem vaccine: What about ADE in the case of this vaccines? Authors mentioned that this type of vaccines have been evaluated in LAIV and IIV platforms. Any difference in the outcome.

  1. ADE is discussed in the context of H2N2 vaccination of pigs which was attributed to HA-2 specific antibodies.

We could not find any studies indicating that LAIV may cause ADE in animal models. Added the following section to address the question – “It has been previously theorised that sub-neutralising antibody levels and activation of FcγRI and FcγRIIA could promote viral cell entry, replication and subsequent antigen presentation and immunogenicity of live attenuated viral vaccines [118].  However, a recent review has noted that in the context of influenza, LAIVs have been repeatedly shown to be safe and effective in animal models with no indications of VAERD or ADE [119].”

In general, more figures should be added to exemplify the information provided, which would make the review more valuable. At least, a table summarizing the different strategies discussed, pros and cons and state of evaluation in clinical trials (if it is possible should be provided). Information about the different platforms and animal models would be informative. Figure 1 is hard to understand, and no nomenclature is provided in the figure legend.

  • We agree a table would benefit the review as a useful resource and is added below. The legend of Figure 1 has also been expanded and details of the figure expanded in the final discussion.
  • The clinicaltrials.gov has over 2,075 influenza vaccine trials in progress. A previous review from our group in 2018 summarised the number of trials in progress (PMID: 30013557), and the discussion in this review now includes the halted MVA/NP+M1 and Chimeric HA LAIV trials.

Table 1: Summary of potential outcomes and hurdles for next generation influenza vaccines

Vaccine type

Example vaccine

Baseline immunity

Hypothetical hurdle

Impact

Potential outcomes

Reference

bnAbs HA/NA vaccine

(H1N1)

Chimeric HA LAIV + AS03

(Phase I trial halted)

H1N1 primed

New H1N1 pdm virus

Enhanced immunity

Broad heterosubtypic protection

[93–95]

H3N2 primed

Reduced immunity

Impartial protection

[26,27]

T cell peptide vaccine

(H5N1)

FLU-V

(Phase IIb trial ongoing)

H1N1 primed

Reduced immunity

Impartial protection

[37,42]

H1N1 primed

H5N1 pdm virus

Enhanced immunity

Broad heterosubtypic protection

[62,70]

Rare HLA-type

Lack of T cell priming

Reduced immunity

Impartial protection

[47,48]

Adenovirus vectored vaccine

ChAdOx1 NP + M1

(Phase I trial completed)

Preexposure to vector adenovirus

Vector Immunity disrupting priming

Reduced Immunity

Impartial protection

[85,87]

Naïve to adenovirus vector

No pre-exposure

Enhanced Immunity

Broad heterosubtypic protection

[72,74,91]

Why neuraminidase (NA)-based vaccines were not discussed?

Section 6 has been added to include a discussion of NA immunity.

Minor comments:

Line 32: Please remove ‘’…threat with higher mortality rate...’’ -> removed

Line 33: replace term accrue -> acquire

Line 44: Reference is missing -> updated

Line 116: Please replace utilization -> priming or boosting

Line 244-245: References are missing -> 5 references are given within this sentence and it unclear on further references needed.

Line 327: It gives the impression that none of the vaccines discussed have reached approval which is not true. Please rephrase.

  1. Next generation vaccines have proven safety and immunogenicity, but do not have licensure approval. This sentence has been removed.

Line 286-294 This paragraph seems to be out of place. I would recommend reorganize the text

  1. The paragraph has been removed to streamline the review.

Round 2

Reviewer 2 Report

The revised version of this article review has improved considerably from the previous version. Authors has done a satisfactory work addressing most of my comments. Below I have few suggestions that authors should address before article review is accepted for publication

  1. In sentence:

…‘’However, in a small study of Flumist in South Africa over 60-year-olds, there is no increased rate of protection from acquisition of infection or duration of immunity for LAIV compared to IIV, however low incidence of infection may preclude solid evidence [30]’’ …. Please also state at the end that age could be a factor that preclude the evidence

  1. mRNAs vaccines discussion was improved but it is still not enough. More information must be provided about the Influenza mRNA vaccines under evaluation. Some references that should be evaluated for suitability (I apologize if some have been already included). Brief explanation of the work in each article should be incorporated (at least for some of them)

: Wong SS, Webby RJ. An mRNA vaccine for influenza. Nat Biotechnol. 2012

Dec;30(12):1202-4. doi: 10.1038/nbt.2439. PMID: 23222788.

2: Freyn AW, Ramos da Silva J, Rosado VC, Bliss CM, Pine M, Mui BL, Tam YK,

Madden TD, de Souza Ferreira LC, Weissman D, Krammer F, Coughlan L, Palese P,

Pardi N, Nachbagauer R. A Multi-Targeting, Nucleoside-Modified mRNA Influenza

Virus Vaccine Provides Broad Protection in Mice. Mol Ther. 2020 Jul

8;28(7):1569-1584. doi: 10.1016/j.ymthe.2020.04.018. Epub 2020 Apr 19. PMID:

32359470; PMCID: PMC7335735.

3: Petsch B, Schnee M, Vogel AB, Lange E, Hoffmann B, Voss D, Schlake T, Thess

A, Kallen KJ, Stitz L, Kramps T. Protective efficacy of in vitro synthesized,

specific mRNA vaccines against influenza A virus infection. Nat Biotechnol. 2012

Dec;30(12):1210-6. doi: 10.1038/nbt.2436. Epub 2012 Nov 25. PMID: 23159882.

4: Zhuang X, Qi Y, Wang M, Yu N, Nan F, Zhang H, Tian M, Li C, Lu H, Jin N. mRNA

Vaccines Encoding the HA Protein of Influenza A H1N1 Virus Delivered by Cationic

Lipid Nanoparticles Induce Protective Immune Responses in Mice. Vaccines

(Basel). 2020 Mar 10;8(1):123. doi: 10.3390/vaccines8010123. PMID: 32164372;

PMCID: PMC7157730.

5: Vogel AB, Lambert L, Kinnear E, Busse D, Erbar S, Reuter KC, Wicke L,

Perkovic M, Beissert T, Haas H, Reece ST, Sahin U, Tregoning JS. Self-Amplifying

RNA Vaccines Give Equivalent Protection against Influenza to mRNA Vaccines but

at Much Lower Doses. Mol Ther. 2018 Feb 7;26(2):446-455. doi:

10.1016/j.ymthe.2017.11.017. Epub 2017 Dec 5. PMID: 29275847; PMCID: PMC5835025.

6: Feldman RA, Fuhr R, Smolenov I, Mick Ribeiro A, Panther L, Watson M, Senn JJ,

Smith M, Almarsson Ó¦, Pujar HS, Laska ME, Thompson J, Zaks T, Ciaramella G. mRNA

vaccines against H10N8 and H7N9 influenza viruses of pandemic potential are

immunogenic and well tolerated in healthy adults in phase 1 randomized clinical

trials. Vaccine. 2019 May 31;37(25):3326-3334. doi:

10.1016/j.vaccine.2019.04.074. Epub 2019 May 10. PMID: 31079849.

7: Bahl K, Senn JJ, Yuzhakov O, Bulychev A, Brito LA, Hassett KJ, Laska ME,

Smith M, Almarsson Ö, Thompson J, Ribeiro AM, Watson M, Zaks T, Ciaramella G.

Preclinical and Clinical Demonstration of Immunogenicity by mRNA Vaccines

against H10N8 and H7N9 Influenza Viruses. Mol Ther. 2017 Jun 7;25(6):1316-1327.

doi: 10.1016/j.ymthe.2017.03.035. Epub 2017 Apr 27. PMID: 28457665; PMCID:

PMC5475249.

  1. The NA section has improved considerably the quality of the review. If it is possible, more references should be added. Additionally, in sentence:

…’’NAIs are the only clinically available antiviral drugs to treat influenza and antiviral resistance has already been observed in circulating IAV [108–110]’’… Please double check and rephrase since Baloxavir is also clinically available

Author Response

Reviewer 2, round 2

The revised version of this article review has improved considerably from the previous version. Authors has done a satisfactory work addressing most of my comments. Below I have few suggestions that authors should address before article review is accepted for publication

  1. We thank the reviewer for their positive reply and many constructive comments below which have been incorporated to improve our review.
  1. In sentence:

…‘’However, in a small study of Flumist in South Africa over 60-year-olds, there is no increased rate of protection from acquisition of infection or duration of immunity for LAIV compared to IIV, however low incidence of infection may preclude solid evidence [30]’’ …. Please also state at the end that age could be a factor that preclude the evidence

  1. Updated to:

“However, in a small study of Flumist in South Africa over 60-year-olds, there is no increased rate of protection from acquisition of infection or duration of immunity for LAIV compared to IIV. Though the age of study participants (60-95 years old) which are not the target population for LAIV as it is typically precluded in over 65 years and over and low incidence of infection may preclude these findings [30].”

  1. mRNAs vaccines discussion was improved but it is still not enough. More information must be provided about the Influenza mRNA vaccines under evaluation. Some references that should be evaluated for suitability (I apologize if some have been already included). Brief explanation of the work in each article should be incorporated (at least for some of them)
  1. We thank the reviewer for this thorough list of work on mRNA vaccines and have incorporated an updated paragraph to describe their use for universal influenza vaccines.

“The COVID19 pandemic has demonstrated the high efficacy of mRNA-based vaccines compared to other vaccine formats. mRNA-based influenza vaccines have also long been in development and may be key to universal vaccine development. mRNA vaccines have an advantage of being self-adjuvating for TLR 3, 7 and 8, whilst incorporating nucleoside modifications [101]avoids excess inflammation through the TLR system. This also increases protein production [102] and the mRNA lipid nanoparticle vaccine formulation improves delivery and stability [103,104]. Vaccine formulations including HA of pandemic potential [103] and combination strategies of HA, NA, NP [105] and M2e [101] have shown increased protection in comparison to standard of care inactivated vaccines across multiple species [103,105]. Human trials include HA pandemic potential IAV subtypes [106] and seasonal QIV HA based formulations [107]. The combination of influenza QIV and COVID19 vaccines by Moderna [107] in an ongoing COVID19 vaccinated phase I/II trial will establish the ability to boost prior immunity with mRNA vaccines. Whilst prime boost dosing appears to increase immunogenicity [101,105], dose sparing may also be possible for doses [108]. Whilst the mRNA strategy is clearly immunogenic, HA only platforms will only address single pandemic potential viruses like recombinant HA vaccines, therefore employing multivalent antigen combination approaches or mosaic HA within mRNA vaccines holds the most promise.

  1. Wong SS, Webby RJ. An mRNA vaccine for influenza. Nat Biotechnol. 2012 Dec;30(12):1202-4. doi: 10.1038/nbt.2439. PMID: 23222788.
  2. Freyn AW, Ramos da Silva J, Rosado VC, Bliss CM, Pine M, Mui BL, Tam YK, Madden TD, de Souza Ferreira LC, Weissman D, Krammer F, Coughlan L, Palese P, Pardi N, Nachbagauer R. A Multi-Targeting, Nucleoside-Modified mRNA Influenza Virus Vaccine Provides Broad Protection in Mice. Mol Ther. 2020 Jul 8;28(7):1569-1584. doi: 10.1016/j.ymthe.2020.04.018. Epub 2020 Apr 19. PMID: 32359470; PMCID: PMC7335735.
  3. Petsch B, Schnee M, Vogel AB, Lange E, Hoffmann B, Voss D, Schlake T, Thess A, Kallen KJ, Stitz L, Kramps T. Protective efficacy of in vitro synthesized, specific mRNA vaccines against influenza A virus infection. Nat Biotechnol. 2012 Dec;30(12):1210-6. doi: 10.1038/nbt.2436. Epub 2012 Nov 25. PMID: 23159882.
  4. Zhuang X, Qi Y, Wang M, Yu N, Nan F, Zhang H, Tian M, Li C, Lu H, Jin N. mRNA Vaccines Encoding the HA Protein of Influenza A H1N1 Virus Delivered by Cationic Lipid Nanoparticles Induce Protective Immune Responses in Mice. Vaccines (Basel). 2020 Mar 10;8(1):123. doi: 10.3390/vaccines8010123. PMID: 32164372; PMCID: PMC7157730.
  5. Vogel AB, Lambert L, Kinnear E, Busse D, Erbar S, Reuter KC, Wicke L, Perkovic M, Beissert T, Haas H, Reece ST, Sahin U, Tregoning JS. Self-Amplifying RNA Vaccines Give Equivalent Protection against Influenza to mRNA Vaccines but at Much Lower Doses. Mol Ther. 2018 Feb 7;26(2):446-455. doi: 10.1016/j.ymthe.2017.11.017. Epub 2017 Dec 5. PMID: 29275847; PMCID: PMC5835025.
  6. Feldman RA, Fuhr R, Smolenov I, Mick Ribeiro A, Panther L, Watson M, Senn JJ, Smith M, Almarsson Ó¦, Pujar HS, Laska ME, Thompson J, Zaks T, Ciaramella G. mRNA vaccines against H10N8 and H7N9 influenza viruses of pandemic potential are immunogenic and well tolerated in healthy adults in phase 1 randomized clinical trials. Vaccine. 2019 May 31;37(25):3326-3334. doi: 10.1016/j.vaccine.2019.04.074. Epub 2019 May 10. PMID: 31079849.
  7. Bahl K, Senn JJ, Yuzhakov O, Bulychev A, Brito LA, Hassett KJ, Laska ME, Smith M, Almarsson Ö, Thompson J, Ribeiro AM, Watson M, Zaks T, Ciaramella G. Preclinical and Clinical Demonstration of Immunogenicity by mRNA Vaccines against H10N8 and H7N9 Influenza Viruses. Mol Ther. 2017 Jun 7;25(6):1316-1327. doi: 10.1016/j.ymthe.2017.03.035. Epub 2017 Apr 27. PMID: 28457665; PMCID:PMC5475249.
  1. The NA section has improved considerably the quality of the review. If it is possible, more references should be added. Additionally, in sentence:

…’’NAIs are the only clinically available antiviral drugs to treat influenza and antiviral resistance has already been observed in circulating IAV [108–110]’’… Please double check and rephrase since Baloxavir is also clinically available

  1. Revised to “NAIs are a common clinically available antiviral drugs”.

We appreciate the field targeting NA through vaccines and antivirals, this section already includes 14 references therefore we have not expanded this section but at the start of the section directed readers to a NA focussed review (Krammer et al mBio 2018).
